# Fluid Sensing Using Quartz Tuning Forks—Measurement Technology and Applications

**DOI:** 10.3390/s19102336

**Published:** 2019-05-21

**Authors:** Thomas Voglhuber-Brunnmaier, Alexander O. Niedermayer, Friedrich Feichtinger, Bernhard Jakoby

**Affiliations:** 1Institute for Microelectronics and Microsensors, Johannes Kepler University, 4040 Linz, Austria; friedrich.feichtinger@jku.at (F.F.); Bernhard.jakoby@jku.at (B.J.); 2Micro Resonant OG, 4020 Linz, Austria; a.niedermayer@micro-resonant.at

**Keywords:** viscosity, density, condition monitoring, measurement instrument

## Abstract

We provide an overview of recent achievements using quartz tuning forks for sensing liquid viscosity and density. The benefits of using quartz crystal tuning forks (QTFs) over other sensors are discussed on the basis of physical arguments and issues arising in real world applications. The path to highly accurate and robust measurement systems is described and a recently devised system considering these findings is presented. The performance of the system is analyzed for applications such as the mixing ratio measurement of fuels, diesel-soot contamination for engine oil condition monitoring, and particle size characterization in suspensions. It is concluded that using properly designed systems enables a variety of applications in industry and research.

## 1. Introduction

Piezoelectric sensors can be used to measure mechanical, electrical, optical and thermal parameters [1,2,3,4,5,6,7]. Through manifold surface functionalizations, these sensors are also widely used in chemistry and biology [8,9]. When Giessibl [10] introduced quartz crystal tuning forks (QTFs) for atomic force microscopy (AFM) in 1998, it triggered vivid and ongoing interest in QTFs as sensing elements. The typical dimensions and the fundamental mode shape of such a tuning fork are shown in Figure 1. In Jakoby [11], a wide range of sensors for the dynamic shear viscosity η and the density ρ are reviewed, but the utilization of QTFs is not covered. Due to several advantages of using QTFs for fluid sensing, sophisticated setups have been proposed e.g., by [12,13] prior to our work. With this work, we extend the discussions from [11] to QTFs and compile the key findings of our own work from [14,15,16,17,18,19,20,21,22] into one consistent manuscript and demonstrate that accurately measured fluid parameters facilitate numerous applications, some of which are demonstrated in this paper.

### 1.1. Benefits of Using Tuning Fork Sensors for Fluid Sensing

In order to understand the whole range of benefits of using tuning forks for fluid sensing, they are compared to piezoelectric disc sensors. The first application using a piezoelectric sensor for measuring the physical mass was described by Sauerbrey in 1959 [23], who showed that the resonance frequency $f_{0}$ of thickness shear-vibrating piezoelectric quartz disks reacts very sensitively to a rigid attached mass layer (Δm), following $\Delta f/f_{0}=-\Delta m/m_{0}$ where $m_{0}$ is the resonator mass and $\Delta m\ll m_{0}$. Frequencies can be measured by simple means with highest accuracy and therefore such sensors can be used to measure very small layer thicknesses (nm) in vapor deposition systems, or molecules binding to a functionalized surface [9]. They are known as quartz crystal microbalances (QCM), and can also be used to measure the physical properties of fluids as was first described by Kanazawa and Gordon in 1985 [24]. As may be wrongly suggested by the Sauerbrey equation (Equation (Equation 2)), it is not the static fluid weight which is measured, but instead the dynamic drag of a thin layer of viscous fluid following the small vibrations (<nm) of the surface. If the vibrating surface excites a shear wave, the velocity of a plane shear wave propagating in the *x*-direction decays exponentially with exp(−x/δ), where the characteristic decay length δ is given by
(1)$$\delta =\sqrt{\frac{2\eta }{\rho \omega }},\;\;\;\ldots \text{characteristic decay length of shear waves}$$
with η,ρ, and ω denoting dynamic viscosity, mass-density, and angular frequency. The typical range is from hundreds of nanometers to a few microns.
(2)$$\begin{array}{c} \Delta f=-\frac{2f_{0}^{2}}{A\sqrt{\rho_{p}\mu_{q}}}\Delta m \end{array}$$
(3)$$\begin{array}{c} \Delta f=-f_{0}^{3/2}\sqrt{\frac{\eta \rho }{\pi \rho_{q}\mu_{q}}},\;\;\;Q=\sqrt{\frac{\pi \rho_{q}\mu_{q}}{4\rho \eta f_{0}}} \end{array}$$


In Equations (Equation 2) and (Equation 3) [23,24] the resonance parameter changes for rigid mass and viscous fluid are shown. Here, additional quantities $A,\rho_{q}$, and $\mu_{q}$ are disc face area, density, and shear modulus of quartz, respectively. As two individual fluid parameters ρ and η are of interest, a second resonance parameter associated with the damping has to be considered. A natural choice is the *Q*-factor, but also the damping ratio $\zeta ={\left(2Q\right)}^{-1}$ could be used. However, as Equation Equation 3 shows, ρ and η always appear as products, and can therefore not be measured individually. This is one fundamental drawback which is solved by using QTFs rather than QCMs [25]. Another feature which makes QTFs particularly attractive and was, e.g., pointed out by Matsiev [26] in 1998, is that operation frequencies are typically much lower. Commercial tuning forks that are used, for example, as clock reference in wrist watches, typically operate at 32.768 kHz and are well suited to fluid sensing, whereas the resonance frequencies of QCMs are typically not lower than 1 MHz. Often, it is preferable [11] to measure viscosity at lower frequencies because, due to the onset of non-Newtonian behavior, high frequency responses are difficult to interpret and are less comparable to results provided by laboratory instruments. The low frequency shear resonators introduced by Reichel in [27] follow this line of argumentation, for instance. We note additional benefits of using lower frequencies which are associated with the larger decay lengths δ (Equation (Equation 1)). For instance, sensors featuring small δ are more likely to suffer from gradual contamination, e.g., with surface active additives, the deposition of particles, or the accumulation of gas micro-bubbles. Certain surface acoustic wave sensors using shear horizontal or Love waves are even worse off in this respect. Here, only a thin surface layer is vibrating and therefore the moved sensor mass $m_{0}$ is very low, which makes them extremely sensitive to fluid changes [28], but also to contamination. Due to the high frequencies of such devices (e.g., 116 MHz in [29] for a Love wave sensor) and their small decay lengths, the shear wave cannot pass through a contamination layer to sense the fluid. Also for fluids with a micro-structure such as suspensions or emulsions [30], larger δ are preferable [11]. Despite being the most important characteristic for describing plane shear waves and thus the fluid structure interaction of QCMs, δ does not fully describe the flow around the prongs of a vibrating tuning fork. Generally, when a convex body is vibrating transversely in liquid, there is also displaced fluid flowing around the body which can be described by potential flow theory. Heinisch [31] has shown by experiments that hydrodynamic fluid forces acting on tuning forks of square and circular cross-sections are very similar. The rectangular cross-section in Figure 1 (i.e., aspect ratio 0.57) can be assumed as square-like, which is substantiated by the tabulated values of hydrodynamic functions given in [32]. We therefore base our arguments on the similarity to circular cross-sections, where the shear velocity of the potential flow contribution decays with with $1/r^{2}$ instead of exp(−x/δ) as would be the case for purely shear vibrating sensors. (It is mentioned for the sake of completeness that while $v_{y}\propto \exp\left(-x/\delta \right)$ describes the decay of plane shear-waves, for cylinders the decay of the azimuthal velocity $v_{\theta }$ in the radial direction *r* follows the real part of modified Bessel functions K, i.e., vθ∝−Re(K0(2jr/δ)+K2(2jr/δ)), with $j=\sqrt{-1}$, as is described in [33]). It can therefore be concluded that the extension of the flow regime is larger, and a wider spatial averaging is achieved, which also reduces vulnerability to surface contamination. This theory is underpinned by the results in Section 3.3, where a QTF is used for characterizing particles in suspension much larger in diameter than δ [21]. The above-mentioned benefits also apply to cantilever sensors which can also be used for viscosity sensing [34,35]. Indeed, tuning fork sensors are often considered as two counteracting cantilevers, as is indicated by the mode shape in Figure 1. Furthermore, the hydrodynamic models derived for cantilevers can be used as well [31]. Due to the high degree of similarity, methods established for cantilevers such as the estimation of mechanical stiffness [36,37] or using the QTF as a mass sensor is possible. However, benefits of using tuning fork sensors remain in the form of the insensitivity of the clamping condition at the base of the QTF, for instance. Due to symmetry, a rigid boundary condition is realized between the two prongs for the symmetric modes which is not the case for simple cantilevers where mounting (also termed *attachment* or *anchor*) losses require significant attention [38]. Commercially available QTFs are also highly developed products, featuring low fabrication tolerances and high temperature stability at low costs.

### 1.2. Viscosity and Density Sensing Using QTFs

As mentioned earlier, viscosity and density can be determined from measured resonance parameters, when using QTFs. This is a direct consequence of the flow field having potential flow and vortex flow components [39]. Although the calculation of the relation between (f,Q) and (ρ,η) is complicated in nature for general vibrating geometries (see, e.g., [32] for a rectangular vibrating cantilever, [33] for vibrating cylinders, and [39] for blades), a general model with adjustable parameters, as described in [40] can be used:
(4)$$\begin{array}{c} \omega_{0}=\frac{1}{\sqrt{m_{0k}+m_{\rho k}\rho +m_{\eta \rho k}\sqrt{\eta \rho /\omega_{0}}}},\;\;\;\;Q=\frac{1}{\omega_{0}}\frac{1}{c_{0k}+c_{\eta k}\eta +c_{\eta \rho k}\sqrt{\omega_{0}\eta \rho }}. \end{array}$$


This model has proven to be useful in practice, because due to modeling errors and fabrication deviations, a model parameter adjustment using test liquids has to be performed anyway in order to achieve accuracies in the range of, e.g., 0.1% in density and 1% in viscosity. The inversion of the model, the choice of suitable reference fluids and the general error propagation are discussed in [15]. The basic findings are that at least three test liquids are required and should be chosen such that they encircle the intended measurement range in the ρ–η plane.

The error propagation from measurement noise to fluid properties is fully predictable and follows the measurement chain depicted in Figure 2. The propagation of measurement noise on current i(t) and voltage V(t) signals to resonance parameters $f_{0}$ and *Q* is considered in [41]. Based on Equation (Equation 4), the propagation of noise to ρ and η can be determined [17,20]. As a rule of thumb, measurement noise provokes relative deviations which are typically more than ten times higher on viscosity than on density.

An additional prerequisite in order to achieve reasonable accuracies is to take into account that viscosity is a highly temperature sensitive quantity. This was firstly cast into an equation by Reynolds [42] in 1886 in the form of $\eta =\eta_{0}\exp\left(-bT\right)$. Since then, many models have been elaborated for different fluid classes, e.g., [43]. A setup with accurate temperature control is presented in Section 2, and the more intricate problems associated with inaccurate temperature control are discussed in more detail.

Only the mechanical fluid parameters density and viscosity have been considered so far. However, when the QTF is immersed in liquid, the electrodes are inevitably in contact with the fluid. The mechanical and electrical influences on the measured electric impedance can approximately be modeled by the equivalent circuit in Figure 3 [44]. To determine the resonance parameters, the admittance spectrum of this circuit is analyzed. However, due to the shunt capacitance and the electrical fluid loading, the maximum admittance amplitude and the −3 dB bandwidth are no characteristics of the mechanical loading but vary with $C_{0},R_{e}$, and $C_{e}$. Figure 4 shows the measured Bode plot in magnitude and phase of the sensor admittance for a silicon oil standard measured at temperatures ranging from 5 °C to 80 °C. The magnitude response is dominated by the effect of the shunt capacitance $C_{0}$. This effect as well as the electrical fluid loading and others, such as drifts of the analog interface electronics, phase shifts and delays, are summarized under the term *background signals*. The aim of a robust resonance estimation algorithm is to eliminate the background signals, leaving only the extended motional branch signal whose Nyquist plot corresponds to a circle. From the fitted RLC elements, the resonance frequency and *Q*-factors are determined by
(5)$$f=\frac{1}{2\pi \sqrt{\left(L_{m}+L_{f}\right)\left(C_{m}+Cf\right)}}\;\;\;\mathrm{and}\;\;\;Q=\frac{1}{R_{m}+R_{f}}\sqrt{\frac{L_{m}+L_{f}}{C_{m}+Cf}}.$$


The significance of a robust estimation method is also illustrated in [45], where the effect of salinity due to a minute amount of dissolving salt on the measured QCM spectra is shown which applies also to QTFs. In the case that the tuning fork is immersed in the conductive fluid, an insulating layer must be provided. This measure prevents electrochemical corrosion, but also excessive conductive currents, which appear as offsets on the sensor signals reducing the signal-to-noise ratio (SNR). In [21], measurements were performed using parylene-C coated tuning forks in aqueous solutions. The achievable SNR in water was not affected compared to non-conductive silicone oils.

As is furthermore shown in Figure 4, the resonance is highly damped at high viscosity (low temperature) and only visible to the naked eye in the phase plot. In general, the measurement uncertainty is higher for more viscous liquids as a result of reduced *Q*-factor due to higher damping. The detection of the resonance frequency is therefore less accurate at lower *Q* in the presence of measurement noise [41]. In order to achieve higher *Q*-factors and SNRs, it would in principle be advisable to use a higher mode of vibration. However, at the second in-plane mode (at ≈190 kHz) spurious other modes (see [46]) interfere, especially when the QTF is immersed in fluids. Therefore, evaluating the fundamental mode allows better overall accuracy. While the density measurement range is virtually not restricted for technically relevant applications, the viscosity range is limited. This is also indicated by the error ellipses shown in Figure 5 for the actual measurement setup presented in Section 2. As is furthermore shown in Figure 4, noise is generally low, even at a low *Q* of 6.5, leaving headroom for measuring much higher viscosities. However, calibrating with certified reference standards (usually silicone oils) is difficult at such high viscosities, because they begin to show viscoelastic behavior at the operating frequencies of the QTF, while their certified value was determined by the manufacturer at steady shear conditions, i.e., zero frequency. It is found that deviations from Newtonian behavior are typically detectable for silicone oil standards with viscosities above 50 mPas. Deviations begin to exceed the tolerance of the certified reference fluids between 100 and 200 mPas. Further details on achievable precision for the setup as used for the applications in Section 3, are reported in [20].

## 2. Measurement System

Our devised measurement system, taking into account the issues discussed above, is shown in Figure 6. It consists of a temperature controlled measuring cell connected to the universal impedance analyzer, both available from Micro Resonant [47]. The impedance analyzer measures the driven resonance parameters of the QTF. Although approaches exist for micro- and nanocantilevers to use the random Brownian motion within the liquid for excitation [34,48], evaluating driven resonances is required for the larger QTFs in order to achieve the required signal-to-noise ratio and to obtain also the phase data. The cell is designed for online oil condition monitoring, which requires detection of tiny variations of fluid parameters in order to monitor oil degradation as close as possible. It has been identified as mandatory [19] to control the temperature of the fluid very accurately. This is predominately due to mismatch of the response times of the used temperature sensor and QTF (*dynamic mismatch*). In principle, in not stabilized conditions, temperature could be measured and the viscosity could be extrapolated to a reference temperature of, e.g., 25 °C when the temperature of the fluid undergoes changes in the interval from 20 °C to 30 °C due to machine cycles or ambient temperature changes. Such devices are commercially available as screw-mount sensors and are therefore convenient for industrial automation, but in application they suffer from dynamic mismatch and ad hoc unknown and varying temperature dependence of the fluid, required for extrapolation. With an active temperature control, this problem is avoided. Due to the low sample volume of the cell of below 1 mL, equilibrium temperature (stable to 10 mK) is reached within 1–2 min. This allows a full characterization of the temperature dependence of the fluid parameters in the range of 0 °C to 100 °C within reasonable time. In Figure 5, the measurement system is calibrated using certified viscosity-density standards. Three fluids have been selected for calibration and an additional eight were used for comparison. Good trueness of the measurement system is demonstrated. On the right side of the figure, typical error ellipses showing three times the standard deviation are depicted for certain reference fluids at different temperatures. Details can be found in [20].

## 3. Applications

In this section, three particular applications are demonstrated using the setup described above and the results are discussed.

### 3.1. Real Time Monitoring of Biodiesel Content

Regional fluctuations in fuel quality due to the variable addition of biodiesel are a major challenge for a modern engine management system optimized for efficient combustion and low pollutant emissions. With the QTF sensor implemented in the viscosity-density cell VDC100, one of the most important parameters for injection, the viscosity of the fuel can be determined in real time and the engine control can be adjusted accordingly. Figure 7 shows the viscosities vs. temperature characteristics for different fuels measured with the system in Figure 6. The viscosities of biodiesel differ clearly from fossil diesel B0 in value and temperature characteristic.
(6)$$\begin{array}{ccc} \ln(\eta ) & =\ln\left(\eta_{0}\right)\left(1-\varphi \right)+\ln\left(\eta_{100}\right)\varphi -\alpha \varphi \left(1-\varphi \right) & \ldots \text{Grunberg-Nissan model} \end{array}$$
(7)$$\begin{array}{ccc} \rho & =\rho_{0}\left(1-\varphi \right)+\rho_{100}\varphi -\beta \varphi \left(1-\varphi \right) & \ldots \text{binary mixture model} \end{array}$$


In Figure 8, viscosities and densities of diesel mixtures are shown for three temperatures and the mixing models given in Equation (Equation 6) and Equation (Equation 7) are plotted as solid lines.

The quantities denoted with subscripts 0 and 100 represent B0 diesel and pure biodiesel (B100), respectively, and φ is the mass fraction of biodiesel ranging from 0 to 1. The logarithm ln of viscosity follows a simple binary mixture model based on the Grunberg–Nissan model [49]. Here, the mass fractions φ are used instead of the usual mole fractions, which gives better agreement in this application. The parameter α accounts for a small curvature and is adjusted for best fit to α = {0.071, 0.069, 0.066} ln(Pas) for the three temperatures *T* = {15, 25, 50} °C, respectively. For density, a similar model applies with curvature parameter β = {3.6, 3.8, 4} kg/m^3^. Figure 8 also shows the results with α=0 and β=0 as dash-dotted lines, which indicate that in the low mass fraction range, reasonably good results without fitting for α and β can be obtained. Figure 9 shows the time series of each measurement consisting of 300 consecutive raw measurements. It is emphasized that these data points are completely independent. No a priori knowledge of previous data is used to calculate the actual data point. This means that there is no filtering or averaging being used, which leaves plenty of headroom for more sophisticated signal processing. Given the low noise of the measured data, the standard deviations of the mass fraction calculated from density ($\sigma_{\varphi ,\rho }$) and viscosity ($\sigma_{\varphi ,\eta }$) are approximately given for the three temperatures in the low mass fraction range of 0 to 10% by
(8)$$\begin{array}{cc} \sigma_{\varphi ,\rho } & \approx \frac{\sigma_{\rho }}{|\rho_{100}-\rho_{0}|}=\left\{101,\;112,\;125\right\}\mathrm{ppm}, \end{array}$$
(9)$$\begin{array}{cc} \sigma_{\varphi ,\eta } & \approx \frac{\sigma_{\eta }}{\eta \left|\ln\left(\frac{\eta_{100}}{\eta_{0}}\right)\right|}=\left\{59,\;59,\;90\right\}\mathrm{ppm}. \end{array}$$


It is observed that the mass fraction can be determined more accurately from viscosity, but the noises are on the same order. Therefore, both results can be averaged using some adequate weighting to further reduce noise. It can be concluded that changes in biodiesel content of 0.1% can be reliably detected out-of-the-box using the measurement setup, with headroom for further optimizations.

### 3.2. Diesel and Soot Determination in Engine Oil

The diesel particle filters in cars require a periodic regeneration to prevent clogging. For the regeneration, late injected and not ignited diesel is blown into the particle filter to burn off accumulated soot. However, this process is one of the main causes for the entry of fuel into the engine oil. Oil dilution occurs when parts of the late injected fuel are wiped into the crankcase by the piston rings and thus into the engine oil [50]. The addition of biodiesel exacerbates this issue, because due to the higher boiling point, evaporation at the cylinder liners is much lower than for B0 diesel. Future plans to increase the biodiesel share in the EU from currently 7% to 10% by 2020 pose a challenge to engine manufacturers. It is therefore of interest to monitor and assess the effectiveness of different injection schemes online without requiring off-site lab analyzes.

In [28], it is shown that artificially fuel-diluted engine oil can be discriminated from used engine oil by simultaneously monitoring viscosity and permittivity. Although the permittivity and conductivity of the fluids can be determined as a byproduct from the background spectra (see Figure 3) when using QTFs, this method is of limited accuracy, because the background signals are composed of various signal components which are not associated with fluid permittivity, for instance, thermal drifts of the analog interface electronics. To achieve the required permittivity sensing accuracy, an additional sensor would be required. In this application, we show that permittivity can be replaced by a density measurement, and the VDC100 setup as described above can be used to calculate diesel and soot contents in engine oils. An essential point in understanding the oil dilution process is that not only diesel dilution occurs but also the soot content increases during engine operation. While diesel and engine oil are very similar in density, diesel entry reduces viscosity significantly. Unfortunately, the steadily rising soot content increases viscosity such that they compensate partly. However, as is shown in Figure 10, the influence of soot and diesel can be separated when the density is measured in addition. For this purpose, a model mapping viscosity and density to diesel and soot similar to Equation (Equation 6) and (Equation 7) is determined. The laboratory test setup shown on the right in Figure 10 is used to demonstrate the ability of the QTF setup for diesel and soot determination. The initial oil in the fluid vessel contains four parts fresh oil and one part used oil with a diesel concentration of 5% (known from lab analysis) giving an overall diesel content of 1%. The soot content is unknown and is defined in units of used-oil equivalents, i.e., 0.2 for the mixture. The built-in IO channels of the MFA200 are used for pump control to sample the fluid reservoir in intervals of 8 min. In the meantime, the sample is tempered and measured giving the measurements in Figure 11. Two sample fluids, used oil (A) and diesel diluted used oil (B), are added alternately at a rate of 100 µL/min to the initial sample volume of 200 mL using a computer controlled syringe pump. The fluid vessel is continuously mixed by a magnetic stirrer to guarantee thorough mixing and avoid sedimentation effects.

The results determined using the linearized mixture model are shown in Figure 12. Due to the deterministic conditions of the experiment, the actual diesel concentrations and soot equivalents are known throughout the whole measurement and are indicted by the red dashed lines at the plateaus. The good agreement suggests that the QTF setup is well suited for the task.

### 3.3. Particle Characterization by Monitored Sedimentation

The application of a QTF for particle size characterization using the setup shown in Figure 13 has recently been demonstrated by the authors in [21]. The key results are repeated here briefly to introduce the reader to a more academic example which exploits the varying sensitivity along the elongated dimension of the QTF and to substantiate the benefits of the QTF flow profile for measuring suspensions. It was shown that the sensitivity function along this dimension follows the squared mode shape of vibration shown in Figure 1 (Right). In the experiment, particles of defined size distribution were mixed with water and stirred to form a homogeneous distribution within the small measurement vial. When the stirrer was switched off, the particles began to sink with terminal velocities determined by their buoyant mass proportional to the square of the particle diameter. On the left side of Figure 14, the recorded resonance frequencies are shown for two PMMA particle classes with nominal diameters of 20 µm and 40 µm. A homogeneous particle distributed is reached after the stirrer is turned on for approximately 100 s. Residual flow vanishes quickly after turn-off and the particles begin to sediment. It was shown that some sort of particle size chromatography (Physically correct would be the term *sedimentation velocity chromatography*) can be performed on the basis of the resonance frequency or the *Q*-factor shifts for known particle mass-density. The method to obtain the data on the right side of Figure 14 is outlined in [21] in detail. To summarize, it was shown that the problem can be formulated as a Fredholm integral equation which represents an ill-posed inverse problem requiring some sort of numerical regularization in order to give stable results. It is also shown that already a simple truncated singular value decomposition yields good results. The solid lines represent the solutions to the inverse problem and the dashed lines are Gaussian profiles with the standard deviations stated in the datasheet of the used PMMA beads. The agreement is promising and the results are consistent when derived from resonance frequency or *Q*-factor. For this application, close monitoring of the resonance parameters is important in order to track the fastest particles. The output sampling rate of the resonance analyzer was set to 2 sample/s. To avoid excessive shunt currents in the water based suspension, a parylene-C covering layer was deposited on the QTF. Water as base fluid has been used to demonstrate the fundamental suitability of the setup for potential gout crystal characterization in saline buffer solutions. As mentioned before, the decay length of shear waves δ in this application is 3.1 µm and therefore much smaller than the diameter of the particles. It can be observed that the 20 µm and 40 µm particles behave identically except for the different sedimentation velocities, which demonstrates that δ is not the determining quantity, as is sometimes suggested in the literature.

## 4. Conclusions

The unique features and benefits of using piezoelectric quartz tuning forks for sensing liquid viscosity and density were discussed. A recent measurement setup was applied to three applications comprising fuel analysis, engine oil dilution measurement, and particle size characterization. The results confirm the suitability of using QTFs for these profound measurement tasks.

## Figures and Tables

**Figure 1 sensors-19-02336-f001:**
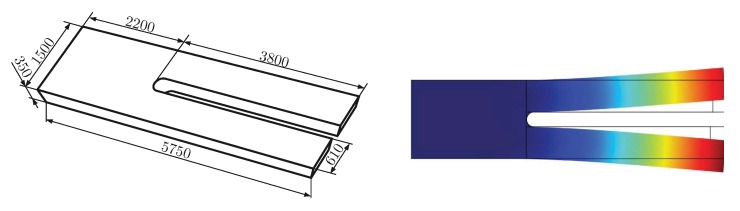
**Left**: Dimensions of the tuning forks (µm) used in the described applications. **Right**: The fundamental in-plane mode shape of the vibration simulated using finite element analysis. The eigenfrequency is 32.7 kHz.

**Figure 2 sensors-19-02336-f002:**
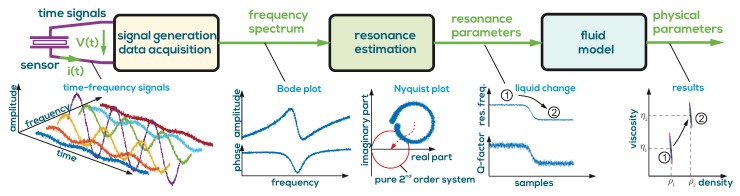
The processing of the data follows the measurement chain. A data acquisition system gives the frequency spectra which are processed by a resonance estimation algorithm. The resonance parameters are related to density and viscosity values ρ and η by a sensor model with predictable error propagation from measurement noise on the time signals.

**Figure 3 sensors-19-02336-f003:**
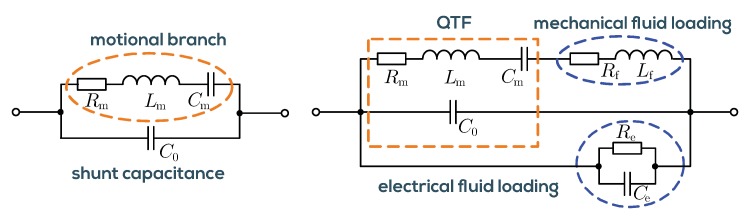
**Left**: Butterworth–Van Dyke equivalent circuit model for the unloaded quartz crystal tuning fork (QTF). The electro-mechanical resonance is modeled by the series resonant circuit termed *motional branch* with shunt capacitance $C_{0}$ in parallel accounting for the capacitance formed by electrodes and quartz material. **Right**: Liquid mass-loading and viscous drag are modeled by $L_{f}$ and $R_{f}$. The conductivity and permittivity of the liquid in contact with the electrodes are modeled by $R_{e}$ and $C_{e}$.

**Figure 4 sensors-19-02336-f004:**
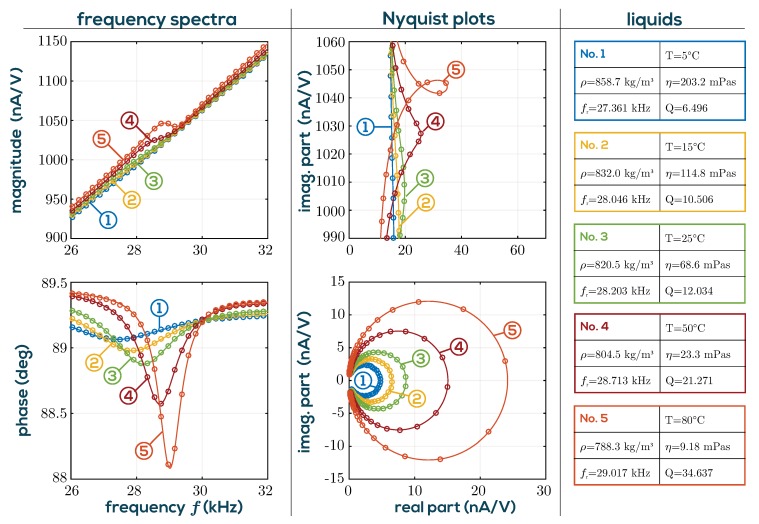
Frequency spectra of measured and fitted admittance signals for a silicone oil at varying liquid temperatures in the range from 5 °C to 80 °C are shown in the left two plots. The signals are dominated by the linear slope of $C_{0}$ in the magnitude, and only the small phase shifts indicate resonances. The above middle figure shows the associated Nyquist plots. In the lower middle figure, the background signals were subtracted, leaving only the motional branch admittance, which resemble circles. The solid lines are results of the resonance estimation algorithm, and the markers show the data points recorded using the system in Figure 6. On the right, the liquids corresponding to the labels (1–5) and the measured resonance parameters are listed.

**Figure 5 sensors-19-02336-f005:**
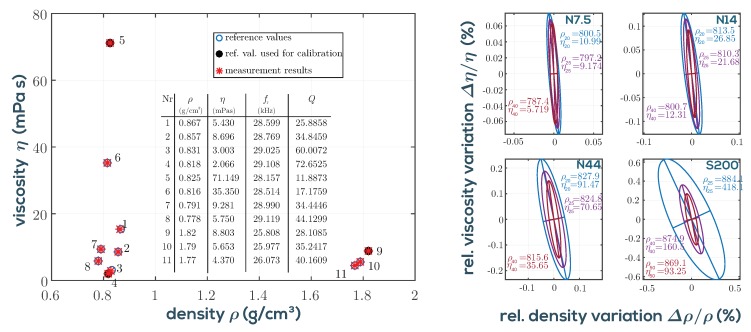
**Left**: The fluid model in Equation (Equation 4) is adjusted using three fluid standards (filled dots No. 4, 5, and 9). The calibration is verified for eight additional fluids. **Right**: Relative deviations due to measurement noise. Error ellipses showing three times the standard deviation are plotted for fluid standards (N7.5, N14, N44, and S200 from Cannon) at three different temperatures each. Units of density and viscosity are kg/m^3^ and mPas, respectively.

**Figure 6 sensors-19-02336-f006:**
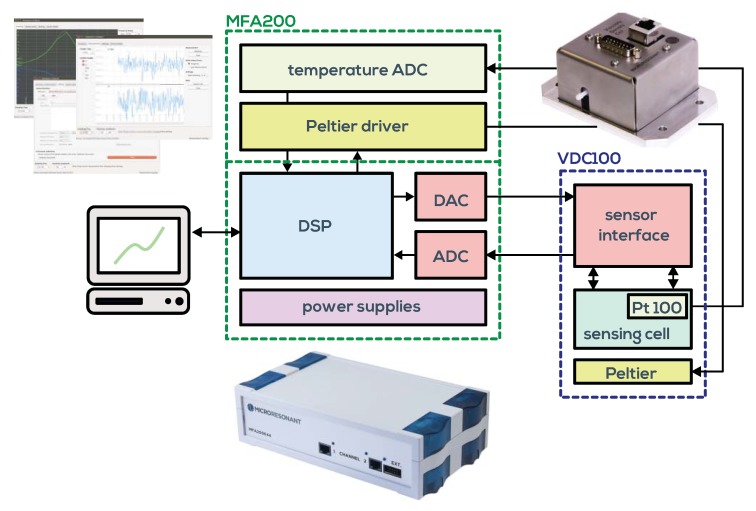
Schematic of the measurement system. The viscosity-density cell VDC100 is connected to the universal resonance analyzer MFA200. The graphical user interface suite of the MFA200 is used for test cycle automation and post processing.

**Figure 7 sensors-19-02336-f007:**
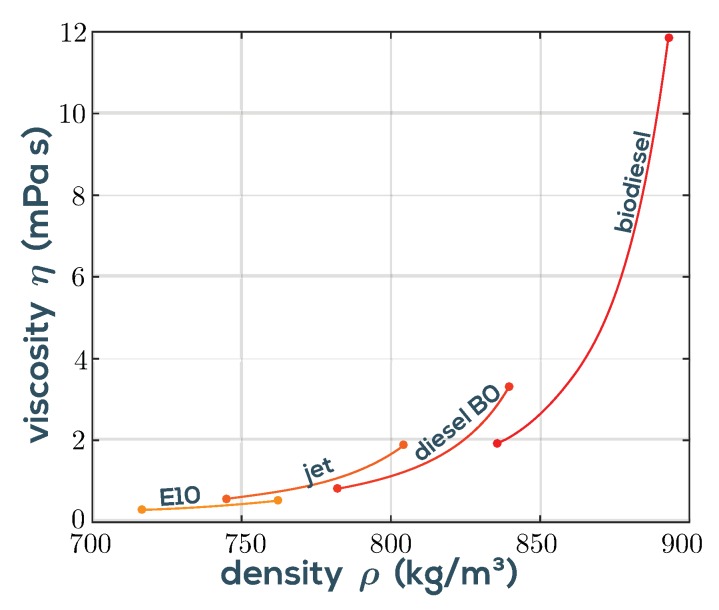
Viscosity-density diagram of fuels in a temperature range of 0–80 °C (E10: 0–40 °C) measured using the VDC100 + MFA200 setup shown in Figure 6. E10 is a fuel mixture of 10% anhydrous ethanol and 90% gasoline and B0 diesel is of fossil origin only. Viscosities are higher at lower temperature.

**Figure 8 sensors-19-02336-f008:**
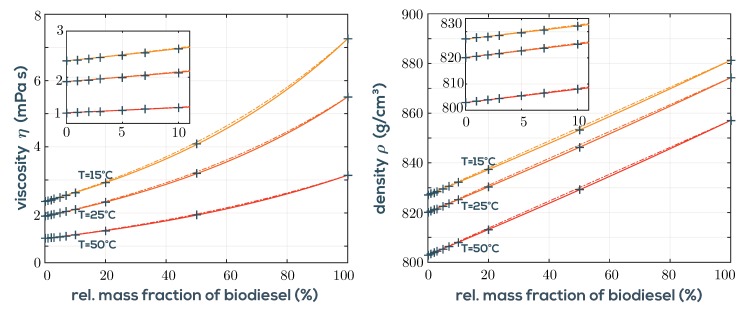
**Left**: viscosities for varying biodiesel content in fossil diesel. The lines follow a Grunberg–Nissan model in Equation (Equation 6). **Right**: Densities for varying biodiesel content in fossil diesel. The lines are given by the binary mixture model in Equation (Equation 7). Dash-dotted lines represent the respective model with fitting parameters α and β set to 0.

**Figure 9 sensors-19-02336-f009:**
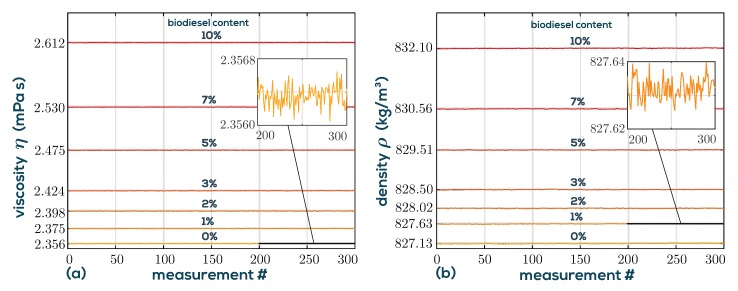
Sample series of viscosity (**a**) and density (**b**) measurements for varying concentrations at a temperature of *T* = 15 °C recorded at a rate of 1 sample/s. Each of the 300 samples is individually determined.

**Figure 10 sensors-19-02336-f010:**
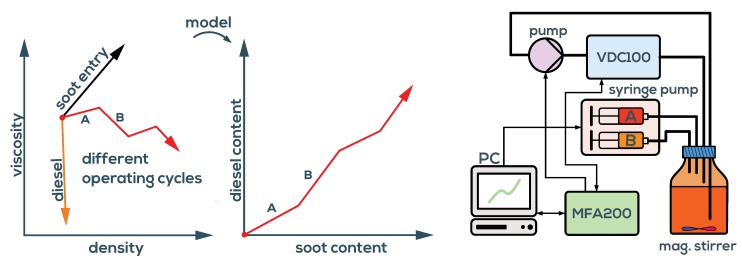
**Left**: diesel and soot entry influence viscosity and density differently and can therefore be calculated using a fluid model. Various operating conditions at the engine test stand can therefore be evaluated online. **Right**: Schematic of the laboratory test setup for concept evaluation.

**Figure 11 sensors-19-02336-f011:**
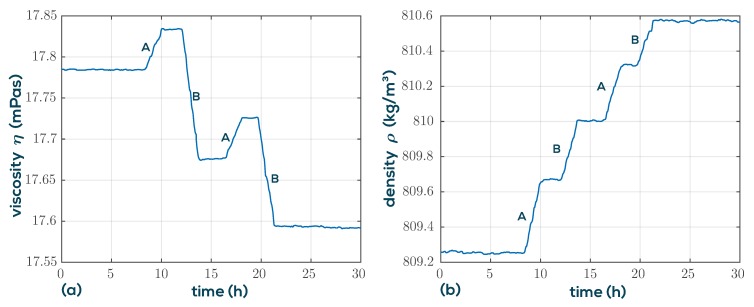
Viscosity (**a**) and density (**(b**) of engine oil for two different dilution scenarios A and B. Soot increases viscosity and density, but diesel reduces viscosity, as shown in Figure 10. Both scenarios introduce diesel and soot, but at different ratios.

**Figure 12 sensors-19-02336-f012:**
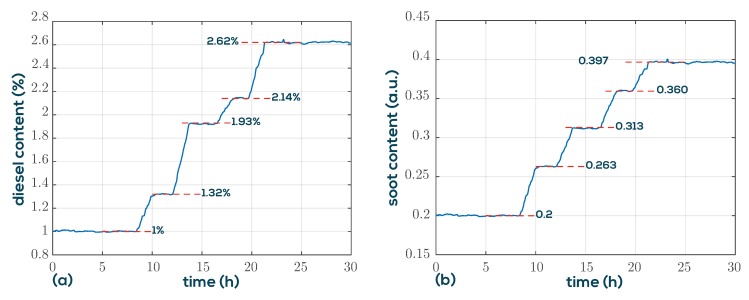
Resulting diesel concentrations (**a**) and soot equivalents (**b**) using a simple linearized mixing model. The dashed lines indicate theoretical values.

**Figure 13 sensors-19-02336-f013:**
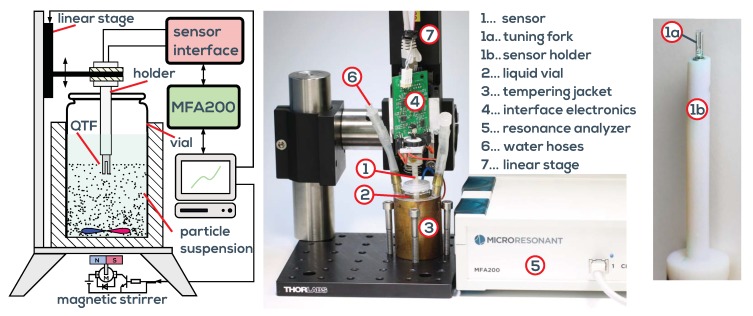
Particles are repeatedly stirred and their sedimentation is monitored using the QTF setup. Temperature is stabilized by means of an externally connected bath circulator.

**Figure 14 sensors-19-02336-f014:**
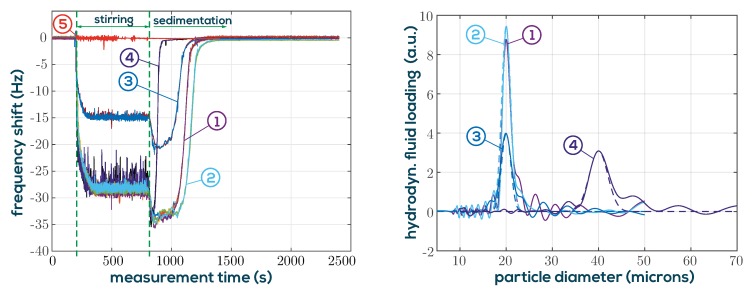
**Left**: The resonance frequencies are measured for a cyclically activated magnetic stirrer. (1) Shows a dispersion of PMMA particles of 20 µm nominal diameter with a concentration of 4.68%. Traces (2) show measurements with the sensor immersed 2.125 mm deeper, which results in a delayed particle front. For trace (3), the concentration is halved at lower immersion. For (4), particles of 40 µm diameter of the same concentration are used. They pass with accordingly higher velocities. (5) shows the measurement using clear DI-water for reference. The noise while stirring is higher due to mechanical vibration. **Right**: Calculated distributions from the resonance frequency characteristics (traces 1–4) in the left figure. The dashed lines shown Gaussian distributions centered at 20 µm and 40 µm with the standard deviation given in the datasheet of the beads.

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
