# Peer review of "Fluid Sensing Using Quartz Tuning Forks—Measurement Technology and Applications"

_sensors, 2019, doi:10.3390/s19102336_

Round 1
Reviewer 1 Report
Dear authors,
Your manuscript is clearly written, the results are quite convincing and I think it will be a valuable work for futher research in fluid sensing with QTFs.
Author Response
Your manuscript is clearly written, the results are quite convincing and I think it will be a valuable work for further research in fluid sensing with QTFs.
Thank you for reviewing and your encouraging comment!
Reviewer 2 Report
The following comments/suggestions should be addressed by the authors
1. Please expand upon the use of QTF sensors in atomic force microscopy. What types of imaging and applications have they had in this field? Have they been used as force sensors to perform force spectroscopy ?
2. Micro-cantilevers, such as AFM cantilevers, and QTF sensors share some of the same properties and therefore the introduction should have a description of how AFM cantilevers have been used as sensors to measure various properties. Indeed they have been used to measure liquid viscosity. The following papers illustrate this and are suggested to cite. It should be noted that it is not compulsory to cite the following papers. They are suggestions.
- Measurement of solution viscosity by atomic force microscopy, N Ahmed, DF Nino, VT Moy - Review of Scientific Instruments, VOLUME 72, NUMBER 6 JUNE 2001
- Viscosity measurements based on experimental investigations of composite cantilever beam eigenfrequencies in viscous media, C Bergaud, L Nicu, Review of Scientific Instruments 71 (6), 2487-2491
3. Viscosity effects and fluid density have also been used to calibrate the stiffness of AFM cantilevers. Could the same theory and experimental methodology developed by Sader et al be applied to QTF sensors ? This should be discussed and the following paper is suggested to cite. It should be noted that it is not compulsory to cite the following paper. It is a suggestion.
- Calibration of rectangular atomic force microscope cantilevers, JE Sader, JWM Chon, P Mulvaney, Review of scientific instruments 70 (10), 3967-3969, 1999
4. Accurate knowledge of the QTF spring constant would seem to be useful as it could allow the QTF to act as a mass sensor much like microcantilevers have been used. Please discuss.
Some mention of methods that could be used to calibrate the spring constant of QTF sensors should be made and citations provided. The papers below are suggested to cite. It should be noted that it is not compulsory to cite the following papers. They are suggestions.
- Atomic force microscope cantilever calibration using a focused ion beam, AD Slattery, JS Quinton, CT Gibson, Nanotechnology 23 (28), 285704 (2012)
- A virtual instrument to standardise the calibration of atomic force microscope cantilevers, JE Sader, R Borgani, CT Gibson, DB Haviland, MJ Higgins, JI Kilpatrick et al. Review of Scientific Instruments 87 (9), 093711, 2016
5. Would there be an advantage to monitoring higher order resonant frequencies for the QTF sensors? The Q factor for higher modes would be larger in air and when immersed in fluid it would still be higher than the corresponding fundamental mode in liquid. This could improve accuracy. Please discuss.
Reviewer 3 Report
The manuscit clearly and comprehensively presents how the quartz tuning fork can measure the viscosity and density of different fluids. Even if the manuscript looks more like a review than a real novelty, which is reflected in the high number of self-citations, the results presented and the accuracy of the measurements are impressive. For this reason I recommend the publication of this paper as it is.
Author Response
The manuscript clearly and comprehensively presents how the quartz tuning fork can measure the viscosity and density of different fluids. Even if the manuscript looks more like a review than a real novelty, which is reflected in the high number of self-citations, the results presented and the accuracy of the measurements are impressive. For this reason, I recommend the publication of this paper as it is.
We thank the Reviewer for the comment! Indeed, we were inconclusive if we should write a review paper or an article. We chose an in-between approach, because there are already very comprehensive reviews, but which lack thorough consideration of the QFT. We wanted to fill that gap as good as possible which however required citations of our own group’s work. We are thankful that the reviewer acknowledges the good results.
Round 2
Reviewer 2 Report
This referee recommends the paper be accepted.